# Retrospective Study of 25 Cases of Acorn Intoxication Colitis in Horses between 2011 and 2018 and Factors Associated with Non-Survival

**DOI:** 10.3390/ani14040599

**Published:** 2024-02-12

**Authors:** Tanguy Hermange, Basile Ruault, Anne Couroucé

**Affiliations:** Unité de Nutrition, PhysioPathologie et Pharmacologie, Ecole Nationale Vétérinaire, Agro-Alimentaire et de L’alimentation Nantes-Atlantique (Oniris), Université Bretagne Loire, 44307 Nantes, France; basileruault@yahoo.fr (B.R.); acourouce@orange.fr (A.C.)

**Keywords:** acorn, colitis, equine, intoxication, prognosis, non-survival

## Abstract

**Simple Summary:**

The aim of this study is to describe acorn intoxication colitis cases in horses and to find variables associated with non-survival. Data from horses presented at an equine hospital from 2011 to 2018 with a final diagnosis of acorn intoxication were included. Diagnosis was based on the following: season, the presence of acorns in the environment, clinical and hemato-biochemical parameters suggestive of a digestive/renal disease, the co-morbidity of companion animals, and post-mortem findings. A total of 25 horses were included. Results suggest that the intoxication may vary from year to year and that the number of cases seems to increase. Clinical signs associated with acorn intoxication were signs of circulatory shock, digestive signs, and abnormal temperature. Several significant clinical pathological findings were also described. Overall, 44% of horses survived. The majority of non-survivors died, or were euthanized, during the first 48 h. The following findings were significantly associated with non-survival: age, hemorrhagic diarrhea, heart rate, hematocrit, creatinine, blood lactate, and thickness of the colon wall at ultrasonography. This study provides equine practitioners with valuable prognostic information in cases of acorn intoxication.

**Abstract:**

The aim of this study is to describe clinical data associated with acorn intoxication and to find variables associated with survival. Data from horses presented at CISCO-ONIRIS from 2011 to 2018 with a diagnosis of acorn intoxication were included. Diagnosis was based on the following: season, the presence of acorns in the environment, clinical and hemato-biochemical parameters suggestive of a digestive/renal disease, the co-morbidity of companion animals, and post-mortem findings. Statistical analysis was completed using Student’s t-test for mean comparisons and a Chi-square test for group comparisons (*p* < 0.05). A total of 25 horses were included, and seasonality suggests that the intoxication may vary from year to year. Clinical signs associated with acorn intoxication were signs of circulatory shock (lethargy, tachycardia, abnormal mucous membrane, tachypnea), digestive signs (diarrhea, ileus, colic), and abnormal temperature. Clinical pathological findings included increased hematocrit, WBC, creatinine, BUN, GGT, AST, CK and decreased albumin. Overall, 44% (11/25) of horses survived. The majority (13/14) of non-survivors died, or were euthanized, during the first 48 h. Findings significantly associated with non-survival were age, heart rate, hemorrhagic diarrhea, ileus, hematocrit, creatinine, blood lactate, and thickness of the colon wall at ultrasonography. This study provides equine practitioners with valuable prognostic information in cases of acorn intoxication.

## 1. Introduction

Acorn intoxication is a well-known concern for ruminants such as cattle and sheep [1], but information in horses is scarce. Cases [2,3,4,5,6,7,8] were reported a few decades ago with limited clinical information. More recently, two case series have been described. The first one is a case review [9] of nine intoxicated horses hospitalized in the UK, and the second one is a post-mortem case review [10] of 19 horses intoxicated and necropsied in France. However, the number of horses is limited in the first study, and clinical information and prognostic factors are not detailed in both studies.

Despite some hypotheses, the pathogenesis of acorn intoxication is not well understood. In other species, the toxicity has been attributed to its richness in gallotannins, especially digallic acid, which is hydrolyzed in the digestive tract, in pyrogallol with hemolytic properties and in gallic acid, which increased vascular permeability [11,12]. Tannins are macromolecules with strong bitterness and astringency properties which can also act with other proteins or enzymes, such as salivary proteins, endothelial cell proteins of the intestinal mucosa, or microbial proteins. Theses mechanisms could cause mucosal lesions, decrease digestion and transit, perturb gut microbiota, and induce renal tubular necrosis [11,13,14].

Toxicity varies between species and individuals. Clinical data are well described in ruminants [1] with gastroenteritis, constipation, and renal tubular necrosis, and other species such as pigs, which seem to be resistant to toxicity. This resistance has been attributed to tannin-binding salivary proteins (TBSPs) which can fix and hydrolyze tannins. Interestingly, an adaptation with inducible TBSPs when animals are exposed to a small account of acorns hulls has been shown in pigs and rodents [14,15]. TBSPs have not been investigated in horses.

Our retrospective study aims to describe the clinical data and to find prognostic factors associated with the non-survival of 25 horses suspected of acorn toxicity hospitalized in the CISCO (Oniris International Center for Equine Health) referral hospital between 2011 and 2018.

## 2. Materials and Methods

### 2.1. Data Collection

Medical records of the CISCO-ONIRIS referral hospital between 2011 and 2018 were reviewed for cases with a final diagnosis of acorn intoxication. Based on another study [9], criteria of inclusion were a final diagnosis of acorn intoxication with 4 of the 6 following criteria: illness during fall, presence of acorn hulls in the environment, co-mortality or co-morbidity of companion animals, acorn hulls in the feces or digestive tract, clinical and hemato-biochemical variables suggestive of a digestive and/or renal disease, and post-mortem findings suggestive of acorn intoxication.

All data obtained from medical records for history, clinical examination, blood analysis, complementary exams, outcome, and post-mortem information were reviewed. History data included seasonality, environment, time of admission, age, sex and breed. Clinical data included rectal temperature, mucous membrane aspect, capillary refill time (CRT), heart rate (HR), respiratory rate (RR), and digestive signs (diarrhea, colic). For statistical evaluation, mucous membrane abnormalities were graded with a score of 0 if normal, a score of 1 for mild abnormalities (dryness, paleness, or congestion alone) and a score of 2 for marked abnormalities (petechia/suffusion, combination of dryness/paleness/congestion, icterus, cyanosis).

Blood analysis included hematology, biochemistry panel (total protein, albumin, urea, creatinine, transaminase (ASAT), gamma glutamyl-transferase (GGT), alkaline phosphatase (ALP), creatine kinase (CK), fibrinogen, and electrolyte (sodium, potassium, chloride, calcium). Other complementary exams included transrectal palpation, naso-gastric intubation, transabdominal ultrasound, and paracentesis.

### 2.2. Statistical Analysis

Normality of continuous data distribution was evaluated using the Shapiro–Wilk W test. If necessary, data were log 10 transformed to normalize distribution. R^®^ software (version 4.0.3) was used to perform the different statistical tests with Student’s t-tests for mean comparisons and Chi-square tests for group (survivals versus non-survivals) comparisons. A *p*-value ≤ 0.05 was considered significant.

## 3. Results

### 3.1. History

Between 2011 and 2018, a total of 25 horses met at least four of the six inclusion criteria with the following distribution: illness during fall (n = 25/25), presence of acorn hulls in the environment (n = 25/25), co-mortality or co-morbidity of companion animals (n = 5/25), acorn hulls in the feces or digestive tract (n = 17/25), clinical and hemato-biochemical variables suggestive of a digestive and/or renal disease (n = 25/25), and post-mortem findings suggestive of acorn intoxication (n = 6/25).

All horses lived in pastures. A marked seasonality was noted with all horses presented between September 19th and October 29th. A large diversity of breeds was represented (seven French warmbloods, three Arabian horses, two Thoroughbreds, eight ponies, three Shetlands, one Welsh X Cob, and one Halfinger) typical of the hospital population. Males were overrepresented with 7 stallions, 10 geldings, and 8 females. However, the proportion of non-survivors in each group was similar (4/7 males, 6/10 geldings and 4/8 females died or euthanized).

The mean age was 16.0 ± 7 years old. Younger age was associated with survival; the survival group (n = 11) age was 12.4 ± 6.0 years old, while the non-survival groups (n = 14) were aged at 18.8 ± 6.6 years old (*p* = 0.01). Furthermore, beyond 13 years, there was a significantly (*p* < 0.05) higher mortality rate (n = 12 non-survivors of 16 horses beyond 13 years old versus 2/9 under 13 years old) with an odds ratio (OR) of 11.0 (Figure 1). The disease duration before admission varied between 4 and 144 h with a mean duration time of 6 h. Duration time of the disease before admission was not associated with survival whatever the period considered: >6 h (n = 12; *p* = 0.82); >12 h (n = 9; *p* = 0.33); >24 h (n = 8; *p* = 0.20).

### 3.2. Clinical Examination

Clinical signs of toxic shock were present with lethargy (25/25), tachycardia (20/23), abnormal mucous membrane (24/25), tachypnea (13/19), and abnormal rectal temperature (4/20). Mean HR at admission was 73 ± 19 bpm with a significant difference (*p* = 0.03) between survivors (63 ± 17 bpm) and non-survivors (80 ± 17 bpm). Considering an HR threshold of 65 bpm, there was a significant difference between groups (*p* = 0.01) with a positive predictive value for non-survival of 79% (Table 1) and an odds ratio of death of 12.8. Marked abnormalities of the mucous membrane (hemorrhagic border, congestion, icterus, cyanosis) versus mild abnormalities (dryness, paleness, or congestion alone) were associated with non-survival (*p* = 0.04). CRT rates higher than 3 s at admission were also associated with non-survival (*p* < 0.05). Two horses presented acute pulmonary edema and died quickly after admission (<3 h) despite intensive care. All the horses having abnormal rectal temperatures at admission (2 horses < 36.5 °C; 2 horses > 38.5 °C) died or were euthanized.

Considering digestive signs, 68% (17/25) showed diarrhea, including 6 with hemorrhagic diarrhea, 28% (7/25) had signs of colic, and 28% (7/25) showed signs of ileus. All horses with hemorrhagic diarrhea died. For the seven horses which presented ileus at admission, five horses stayed in ileus until death and two horses developed diarrhea. Only one of the two horses who developed diarrhea survived. Ileus at admission was associated with non-survival (*p* = 0.05).

### 3.3. Blood Analysis

Hemato-biochemical analysis pointed to dehydration with a combination of increased hematocrit and blood lactate concentration, neutrophil leukocytosis, azotemia with increased urea and creatinine, hypoalbuminemia, and increased hepatic and muscular enzymes (Table 2). Electrolyte disturbances were present in several individuals, but only total calcium was decreased (2.33 ± 0.25 mmol/L). Marked intravascular hemolysis was reported in three horses.

Considering mean values, only hematocrit and blood lactate concentration were significantly different (*p* = 0.01) between survivors and non-survivors. When considering the following thresholds: hematocrit > 60%, blood lactate concentration > 4.5 mmol/L, and creatinine > 229.8 μmol/L, there was a significant difference (*p* = 0.01; *p* = 0.02; *p* = 0.05, respectively) between groups with lower chances of survival and respective odds ratios of 7.5, 6.7, and 5.5 (Figure 2, Figure 3 and Figure 4). Using a threshold of hematocrit > 60%, blood lactate concentration > 4.5 mmol/L, and creatinine > 229.8 μmol/L, the positive predictive value for non-survival was above 90% for each variable (Table 1).

### 3.4. Others Complementary Exams

On rectal palpation, acorn hulls were present in 56% (14/25) of the cases. Thirty-five percent (5/17) of the horses had other abnormalities (three gas distention, one pelvic flexure impaction, one caecal impaction, and one tension band). The nasogastric tube did not yield any reflux for any (0/15) of the intoxicated horses, except for one when the horse was examined by its regular veterinarian before admission.

Thickening of the colon wall was significantly greater (*p* = 0.02) among the horses which did not survive (36 ± 19 mm versus 19 ± 6 mm) (Figure 5). Beyond 22 mm of thickness, the horses presenting such damages of the colon had significantly (*p* = 0.01) lower chances of survive with a positive predictive value of non-survival of 89% (Table 1). Thickening of the small intestine was observed for 35% (6/17) of the horses on which ultrasonography was performed. Abdominal ultrasonography showed severe thickening of the colon wall with values rarely encountered in other diseases with an average of 29 ± 17 mm (Figure 6 and Figure 7).

Abdominal paracentesis was performed on four horses. Modified transudates were present in all horses, and peritoneal lactate concentration was measured. The two non-survivors had very high values (9.4 mmol/L and 10.7 mmol/L) whereas the two survivors had moderately increased values (3.2 mmol/L and 4.9 mmol/L). Unfortunately, the number of horses was not sufficient for statistical analysis.

### 3.5. Outcome

Fifty-six percent (14/25) of the horses died (n = 3) or were euthanized (n = 11), and the overall survival rate was 44%. The length of hospitalization varied from 1 h to 9 days, with a median duration of 48 h. Due to their critical conditions, three horses died or were euthanized in the first 3 h after admission, six others died during the first 24 h, and four more died between 24 and 48 h of hospitalization. Overall, 13/14 of non-survivors died or were euthanized during the first 48 h (Figure 8). Causes of euthanasia included pain non-responsive to treatment, persistence or deterioration of circulatory shock, severe complication such as esophageal lacerations (iatrogenic or secondary to ulcers) or iatrogenic rectal tears (grade 4/4), or respiratory difficulties due to severe pulmonary edema or pleural effusion (diagnosed ante-mortem). The only horse euthanized after 48 h was euthanized because of rectal laceration complications. Three quarters (8/11) of the horses who survived had normal parameters of hydration, proteinemia, or albuminemia after 48 h of hospitalization. Similarly, rapid return of appetite and production of feces during hospitalization seem to be good prognostic factors.

### 3.6. Post-Mortem

Post-mortem findings were available for five horses. A large amount of acorn husks (Figure 9) were observed in the intestinal tract of all horses (n = 5/5). Main lesions observed were an extensive severe edematous colitis in four of the five horses (Figure 10, Figure 11 and Figure 12). The cadaver of the last one was too putrefied for post-mortem examination. Extensive ulcers were present in two horses, while extensive hemorrhages appeared in two others. Other significant lesions included petechia, peritoneal or pleural effusion, renal congestion and adrenal hemorrhage.

## 4. Discussion

This study constitutes a relatively large clinical case review about acorn intoxication in equids and describes prognostic factors to help the equine practitioner assess disease severity. Ideally, definitive diagnosis would have required measurements of tannin metabolites in serum or urine samples. A gas chromatography/mass spectrometry test has been validated for this purpose in cattle [12] and has been used in one horse [2]. However, the test has not been validated for horses and is not routinely available in France. For these reasons and due to the retrospective nature of this study, analyses were not performed. Therefore, several inclusion criteria were chosen based on previous studies [2,9]. All horses met the three main criteria: illness during fall, presence of acorn hulls in the environment, and clinical and hemato-biochemical variables suggestive of a digestive and/or renal disease. The other criteria (presence of acorn in digestive tract, co-morbidity, and post-mortem findings) were not present in all horses. Due to a presumptive diagnosis and cost reasons, investigation of enteropathogens was not performed, either, but it should have been performed to exclude potential bias.

The annual variations observed and the increasing number of cases over the period from 2011 to 2018 are remarkable. The annual variations of intoxication may be due to several factors: acorn quantity, the presence of immature acorns, or acorn tannin concentrations. Data presented to the 2019 Havemeyer Workshop on acute colitis revealed that 78% (15/19) of owners reported that there was an unusual high quantity of acorns hulls when the horses became sick [16]. Sixty-three percent (12/19) thought they looked unusually green/immature, suggesting a role of the quantity and of the aspect of the acorn hulls [16]. The ‘mast effect’ is a well-known phenomenon for botanists. It characterizes the variations of acorn production depending on years with high production every 2 to 5 years depending on oak species and meteorological conditions [17,18,19,20,21]. Moreover, it has been shown that tannins and phenolic acid concentrations vary between oak species [22]. Interestingly, some owners reported that some intoxicated horses lived in their pasture for years with an access and consumption of acorn hulls some years. These horses only developed intoxication recently, suggesting that tannins type or concentrations may vary over time. Also, the increasing number of cases over time in this study, with the two others recent case reviews [19,20], may suggest a new emerging form of acorn intoxication. Although underreported cases may also remain a hypothesis, the absence of cases in the database of our hospital and necropsy unit for decades make it unlikely.

From a clinical point of view, our study did not reveal any type, breed or sex predisposition. A marked seasonality was observed with all cases admitted from September 19 to October 29, which may help the practitioner pay attention to acorn intoxication during this period in order to prevent and/or diagnose the disease. The mean age was 16.0 ± 7 years old, which concurs with the two other reviews [9,10]. Older age correlates with non-survival with an odds ratio of 11.0 beyond 13 years old. Case analysis revealed that the critical clinical status was the cause of euthanasia in all horses, excluding a potential financial bias. Reasons for this older age susceptibility remain unexplained. In human septic shock, older age has been shown to be a negative prognostic factor, and poorer physiological adaptation to circulatory shock is suspected [23]. To our knowledge, a similar phenomenon has not been investigated in horses.

In addition to the marked seasonality, the clinical presentation was quite typical with an acute to fulminant severe edematous colitis leading to toxic shock, which was complicated in several cases with marked azotemia, increase hepatic or muscle enzymes, and intravascular hemolysis. Although not fully detailed in other studies, the hemato-biochemical results were similar to previous reports [2,9] and can be easily understood in view of the acute colitis shock. Abdominal ultrasonography examination revealed marked thickening of the large colon wall, especially in non-survivors with values rarely encountered in other diseases. The thickness was so severe that clear measurement was sometimes difficult (Figure 7). This reflects the severe colon edema as it could be observed post-mortem (Figure 11 and Figure 12).

Older age, signs of shock (HR, hematocrit, blood lactates, blood creatinine), hemorrhagic diarrhea, ileus, and increased colon wall thickness at the transabdominal ultrasonography exam were associated with non-survival and may help practitioners assess prognosis. Using a threshold of hematocrit > 60%, blood lactate > 4.5 mmol/L, creatinine > 229.8 μmol/L, and colon wall thickness > 22 mm places the positive predictive value for non-survival above 89% for each variable.

The overall survival rate was low at 44% but a little superior to the 33% survival rate recently described [9]. The first 48 h appear very critical with the large majority of horses (13/14) dying during this period. As seemingly presented in a previous report [9], the improvement of clinical variables during this period may also be considered as a good prognostic factor.

The main limitation of this study includes the relative low number of horses for statistical analysis which precluded multivariable analyses. However, it remains the larger study of acorn intoxication colitis, and several factors were found to be statistically significant in univariate analyses. Other limitations of this study are inherent to its retrospective and clinical aspects, with inter-individual variability regarding the cases’ assessment and management. Indeed, due to the emergency aspect of the intoxication, the first clinical exam and assessment of horses were not performed by the same clinician every time. However, after emergency hours, all the horses were managed by three experienced practitioners, thereby reducing this potential bias. Other limitations include the lack of a definitive diagnosis with tannin metabolites search and the lack of the exclusion of enteropathogens etiology. However, in our practitioner opinion, the complete clinical description of the patients and the focus on four of six inclusion criteria provide a very likely diagnosis of acorn toxicity.

## 5. Conclusions

This study presents a unique large clinical case review of acorn intoxication in horses. It provides valuable clinical data and prognostic information, which could help equine practitioners in diagnostic procedure and prognostic assessment. The intoxication is particularly severe. Despite intensive care, a large proportion of horses died or were euthanized, and findings associated with non-survival were age, heart rate, hemorrhagic diarrhea, ileus, hematocrit, creatinine, blood lactate, and increased colon wall thickness on ultrasonography.

Interestingly, the intoxication presents an apparent increased number of cases over years and an annual variation which may be related to the ‘mast effect’. The study of tannins metabolites in horse’s serum and in acorn hulls—ideally during years with and without intoxication cases—is now necessary to have a better understanding of the epidemiology and physiopathology of the disease.

## Figures and Tables

**Figure 1 animals-14-00599-f001:**
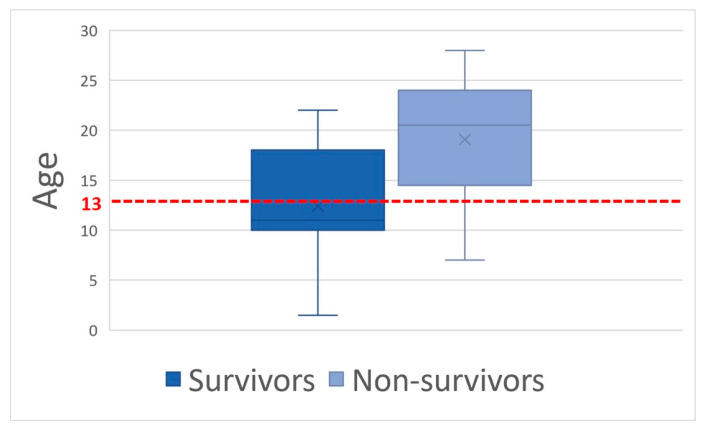
Age box-plot for age between survivors and non-survivors with the 13-year-old threshold in red dotted line.

**Figure 2 animals-14-00599-f002:**
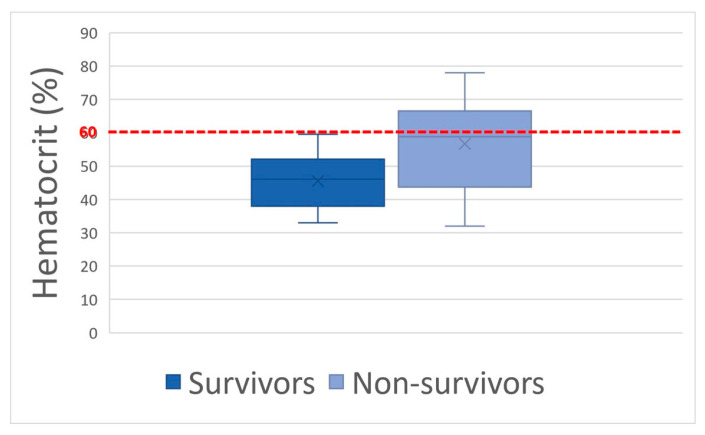
Hematocrit box-plot between survivors and non-survivors with the 60% hematocrit threshold in red dotted line.

**Figure 3 animals-14-00599-f003:**
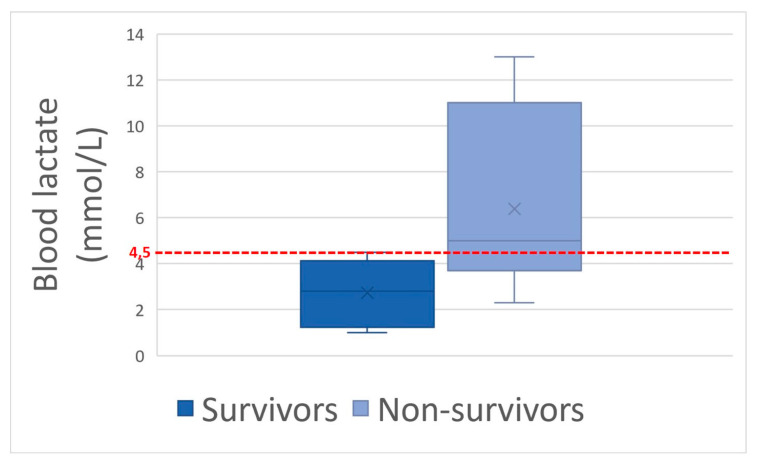
Blood lactates box-plot between survivors and non-survivors with the 4.5 mmol/L lactates threshold in red dotted line.

**Figure 4 animals-14-00599-f004:**
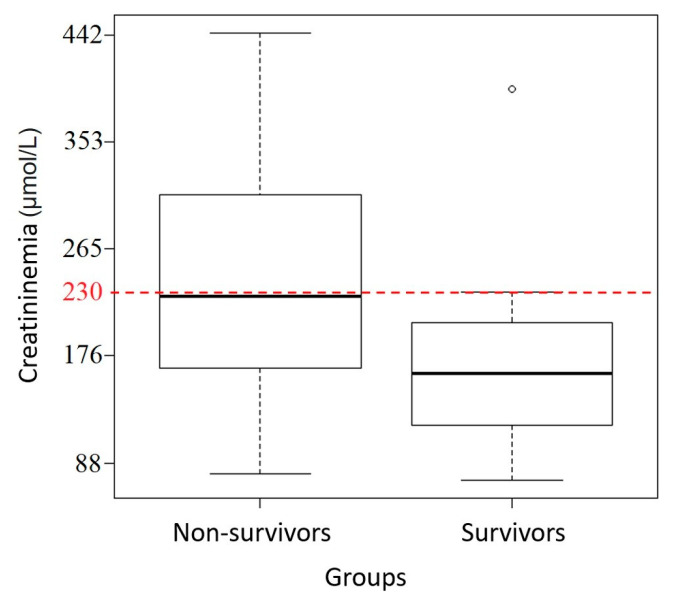
Creatinine box-plot between survivors and non-survivors with the 229.8 μmol/L (26 mg/L) creatinine threshold in red dotted line.

**Figure 5 animals-14-00599-f005:**
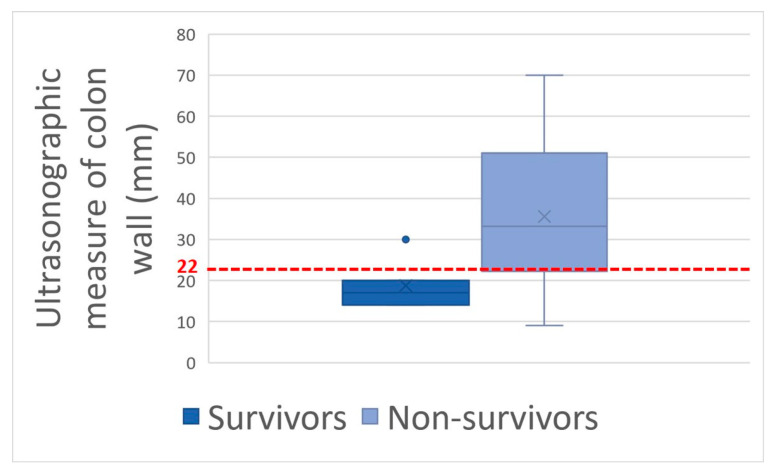
Colon wall thickening box-plot between survivors and non-survivors with the 22 mm thickening value threshold in red dotted line.

**Figure 6 animals-14-00599-f006:**
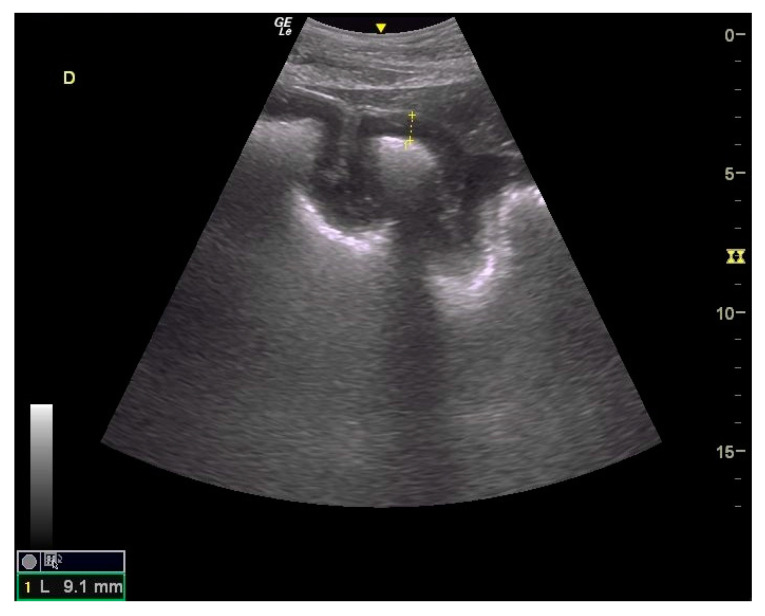
Ultrasound images of large colon. Moderate to marked thickening of large colon with easily visible villosities; the yellow dote line indicates a measurement of the colon wall at 9.1 mm.

**Figure 7 animals-14-00599-f007:**
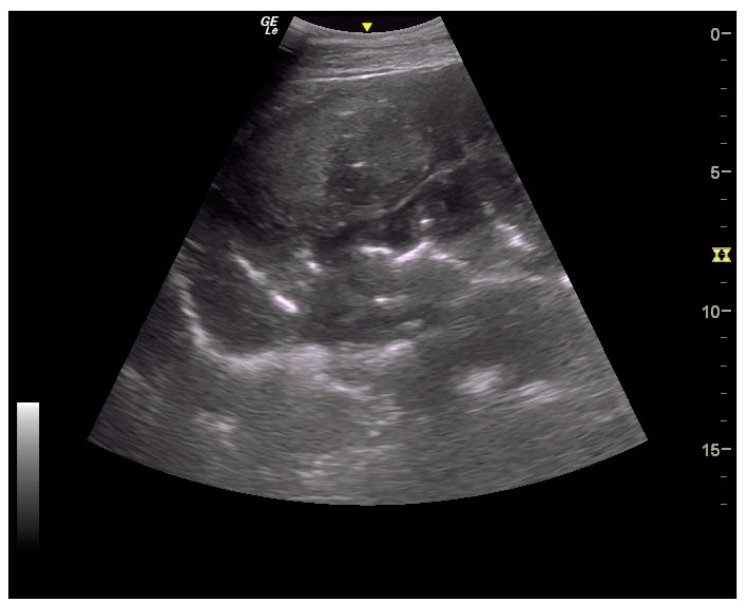
Ultrasound images of large colon. Very severe thickening of large colon.

**Figure 8 animals-14-00599-f008:**
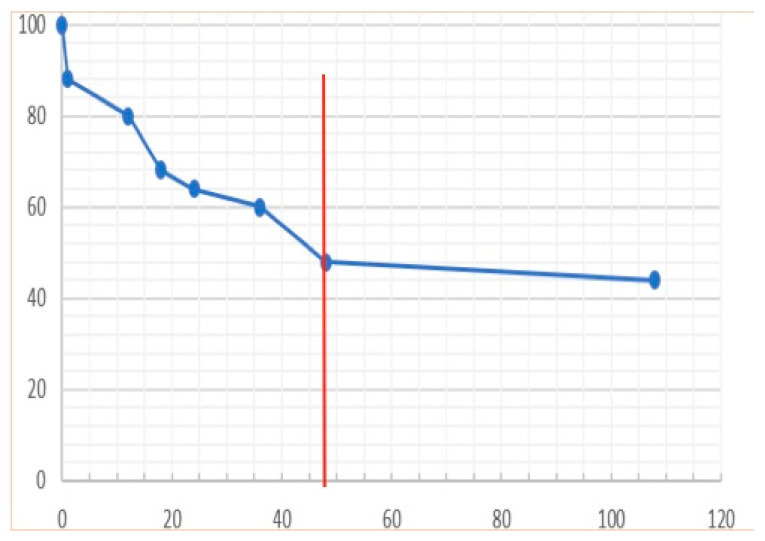
Proportion of horses still alive relative to duration of hospitalization (in hours); the red line marks the threshold of 48 h of hospitalization stay.

**Figure 9 animals-14-00599-f009:**
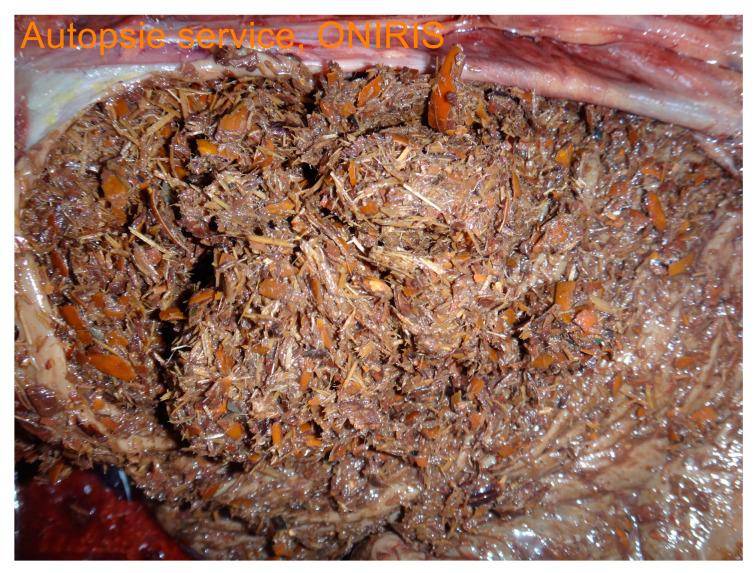
Ascending colon content with numerous acorn hulls.

**Figure 10 animals-14-00599-f010:**
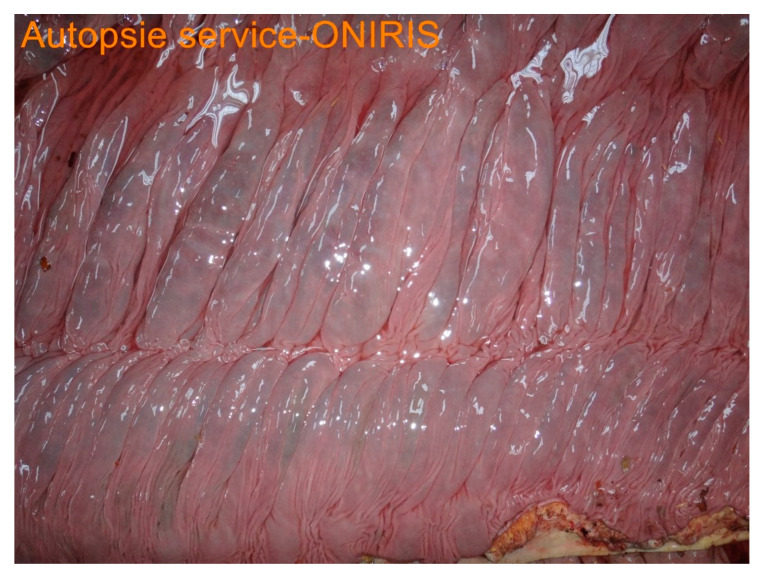
Picture of luminal aspect of ascending colon.

**Figure 11 animals-14-00599-f011:**
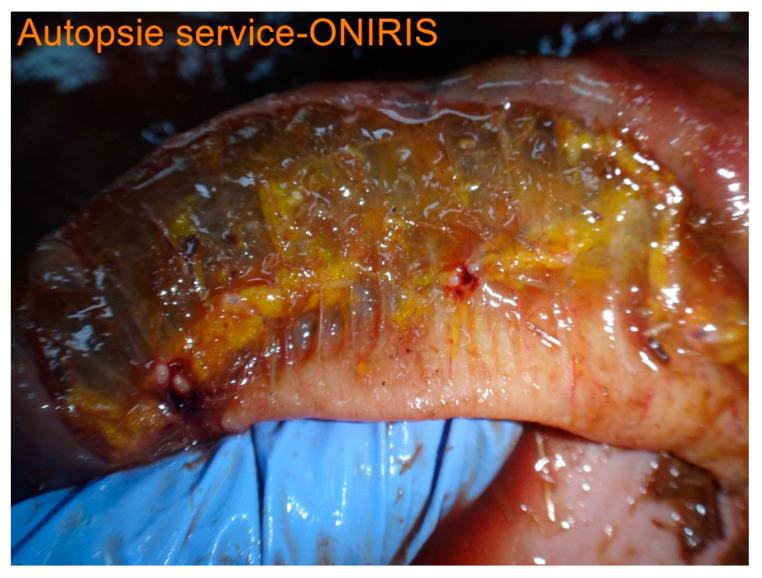
Transversal section of ascending colon.

**Figure 12 animals-14-00599-f012:**
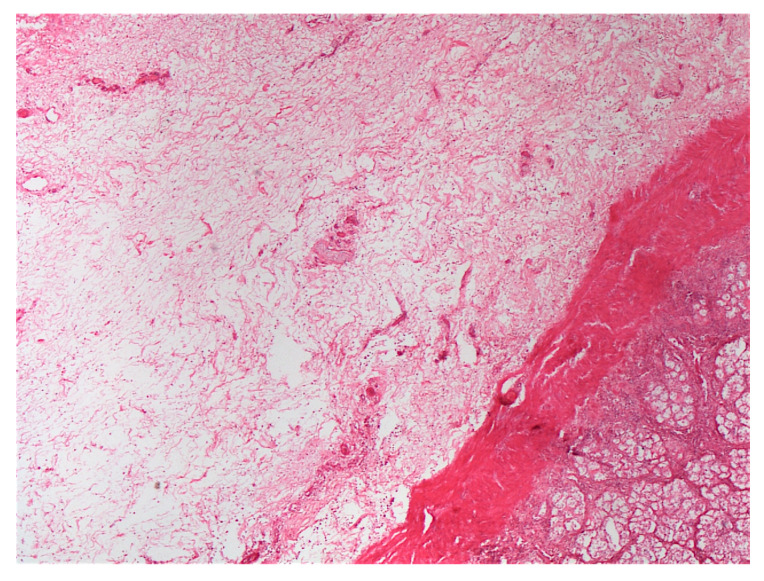
Histologic section of a transversal section; note the severe submucosal edema present in Figure 11 and Figure 12.

**Table 1 animals-14-00599-t001:** Sensitivity, specificity, positive predictive value, negative predictive value of the prognostic factors identified with their respective threshold.

Parameters Thresholds	Sensitivity	Specificity	PPV	NPV	*p*-Value (Chi-Square Test)	Odds Ratios
HR > 65 bpm	85	70	79	57	0.01	12.8
Hematocrit > 60%	64	100	100	56	0.01	7.5
Lactate > 4.5 mmol/L	57	100	100	53	0.02	6.7
Creatinine > 229.8 μmol/L	57	91	89	56	0.05	5.5
Colon wall thickness > 22 mm	80	83	89	59	0.01	10.5

PPV: positive predictive value; NPV: negative predictive value; HR: heart rate; bpm: beats per minute.

**Table 2 animals-14-00599-t002:** Hemato-biochemical results with mean value of total population, survivors and non-survivors.

	Mean Value ± SD	Survivors ± SD	Non-Survivors ± SD	*p*-Value (Student’s *t*-Test)
Hematocrit (%)	52 ± 13	46 ± 9	57 ± 13	*p* = 0.01
Lactates (mmol/L)	4.7 ± 3.6	2.7 ± 1.5	6.4 ± 4	*p* = 0.05
Leucocytes (10^9^/L)	13.7 ± 5.6	13.0 ± 5.5	14.4 ± 5.7	*p* = 0.54
Neutrophils (10^9^/L)	9.6 ± 5.2	9.1 ± 5.6	9.9 ± 5.1	*p* = 0.72
Total proteins (g/L)	62 ± 9	61 ± 10	63 ± 8	*p* = 0.55
Albumin (g/L)	25 ± 5	24 ± 5	25 ± 5	*p* = 0.60
Creatinine (μmol/L)	211.3 ± 100.8	175.0 ± 87.5	239.6 ± 104.3	*p* = 0.11
Urea nitrogen (mmol/L)	22.5 ± 12.9	25.4 ± 13.6	*p* = 0.22	
GGT (μkat/L)	1.12 ± 1.12	1.29 ± 1.65	0.99 ± 0.55	*p* = 0.54
ASAT (μkat/L)	12.71 ± 29.81	19.56 ± 43.80	6.91 ± 4.91	*p* = 0.31
CK (μkat/L)	12.58 ± 9.72	9.59 ± 9.05	13.84 ± 10.10	*p* = 0.32

SD: standard deviation; GGT: gamma-glutamyl-transpeptidase; ASAT: aspartate amino-transferase; CK: creatine kinase.

## Data Availability

The data presented in this study are available on request from the corresponding author (accurately indicate status).

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
