# Peer review of "Retrospective Study of 25 Cases of Acorn Intoxication Colitis in Horses between 2011 and 2018 and Factors Associated with Non-Survival"

_animals, 2024, doi:10.3390/ani14040599_

Round 1
Reviewer 1 Report
Comments and Suggestions for Authors
This is an excellent manuscript which will make a valuable addition to the veterinary toxicological literature. The analysis of prognostic factors is particularly useful, and the photographs and photomicrograph are of excellent quality. There are a few minor typographical errors which I have listed below.
Lines 19 to 20. I recommend revising 'Several clinical pathological findings significantly were also described.' to read 'Several significant clinical pathological findings were also described.'
Line 59. Change 'Clinical data is' to 'Clinical data are'
Line 62. Hyphenate 'tannin-binding'
Line 64. Change 'have been shown' to 'has been shown'. Alternatively, change 'an adaptation' to 'adaptations' in the previous line.
Line 73. Change 'Criteria of inclusion was' to 'Criteria of inclusion were'
Line 117. Change 'odd ratio' to 'odds ratio'
Line 128. Change 'odd ratio' to 'odds ratio'
Table 1. Change 'odd ratio' to 'odds ratio'
Table 1. Change 'thickeness' to 'thickness'
Figure 4 is in a different format compared to Figures 1,2 and 3, and with the positions of the survivor data and the non-survivor data switched. Is it possible to make it more closely resemble Figures 1,2 and 3?
Line 221. I recommend changing 'severe edematous colitis edema' to 'severe edematous colitis'.
Line 261. Remove the comma after 'Although'
Line 264. Change 'point a view' to 'point of view'
Line 269. Change 'odd ratio' to 'odds ratio'
Line 285. I suggest changing 'colon wall thickness' to 'increased colon wall thickness'
Line 301. Delete the comma after 'horses'
Line 310. I suggest changing 'colon wall thickness' to 'increased colon wall thickness'
Comments on the Quality of English Language
The English language is generally of a very high standard, with only a few minor errors that likely reflect the fact that English is presumably a second language for the authors. I have suggested some changes to the grammar and syntax in my comments to the authors.
Author Response
Lines 19 to 20. I recommend revising 'Several clinical pathological findings significantly were also described.' to read 'Several significant clinical pathological findings were also described.'
=> done
Line 59. Change 'Clinical data is' to 'Clinical data are'
=> done
Line 62. Hyphenate 'tannin-binding'
=> done
Line 64. Change 'have been shown' to 'has been shown'. Alternatively, change 'an adaptation' to 'adaptations' in the previous line.
=> done
Line 73. Change 'Criteria of inclusion was' to 'Criteria of inclusion were'
=> done
Line 117. Change 'odd ratio' to 'odds ratio'
=> done
Line 128. Change 'odd ratio' to 'odds ratio'
=> done
Table 1. Change 'odd ratio' to 'odds ratio'
=> done
Table 1. Change 'thickeness' to 'thickness'
=> done
Figure 4 is in a different format compared to Figures 1,2 and 3, and with the positions of the survivor data and the non-survivor data switched. Is it possible to make it more closely resemble Figures 1,2 and 3?
=> I a going to try do do something on it
Line 221. I recommend changing 'severe edematous colitis edema' to 'severe edematous colitis'.
=> done
Line 261. Remove the comma after 'Although'
=> done
Line 264. Change 'point a view' to 'point of view'
=> done
Line 269. Change 'odd ratio' to 'odds ratio'
=> done
Line 285. I suggest changing 'colon wall thickness' to 'increased colon wall thickness'
=> done
Line 301. Delete the comma after 'horses'
=> done
Line 310. I suggest changing 'colon wall thickness' to 'increased colon wall thickness'
=> done
Reviewer 2 Report
Comments and Suggestions for Authors
Depending on the journal requirements - do you capitalize the first word in the title of tables and figures?
Figure above line 121 has no ID or description but appears to be a larger version of Figure 1. Are both figures needed if identical? if so, use two different legends to make them unique.
Below Table 2 has no space between line 153 and 154 text, and table is broken up in this version - so you would need column headers repeated.
Line 185 ... damages of colon ... non-survival of 89% (Table 2). Should be Table 1.
The titles of figures 6 to 12 are a strange presentation. Why not have for example Figure 6. Ultrasound images of large colon. A) Moderate to marked thickening .... . B) very sever thickening.... etc. and Figure 9. Severe edematous colitis A).... B).... C).... and D)... And put a label of A, B, C, etc. on the figures so it can be identified.
Some of the figures had offset margins and split between pages - correct.
Interesting paper and valuable for toxicologists and practitioners.
Comments on the Quality of English Language
Line 177 nasogastric tube did not yielded any reflux..
Line 178 .. except for one when he the horse was examined by..
Line 185 is Table 1
Line 221 were an extensive severe edematous colitis edema in 4 (redundant)
Line 223 ... present in 2 horses while expensive hemorrhages appeared in 2 others. NOT expensive but extensive
Line 230 ... present in Figures 11 et and 12.
Line 233 in equids and describes prognostic factors to help the equine practitioner to assess disease ..
Line 239 ... based on previous studies [1,17] Studies should be plural?
Line 240 .. acorn hulls is in the environment...
Line 243 .. Due to a presumptive diagnosis ...
Line 245 .. have been performed to excluded potential bias.
Line 255 why is there a number 2 superscript after shown... shown2 that tannins and phenolic acid ... What is that in reference to?
Line 295 .. The main limitation ...
Line 297 about of acorn intoxication ...
Line 310 .. thickness of the colon wall at on ultrasonography.
Author Response
Figure above line 121 has no ID or description but appears to be a larger version of Figure 1. Are both figures needed if identical? if so, use two different legends to make them unique.
=> changed
Below Table 2 has no space between line 153 and 154 text, and table is broken up in this version - so you would need column headers repeated.
=> changed
Line 185 ... damages of colon ... non-survival of 89% (Table 2). Should be Table 1.
=> changed
The titles of figures 6 to 12 are a strange presentation. Why not have for example Figure 6. Ultrasound images of large colon. A) Moderate to marked thickening .... . B) very sever thickening.... etc. and Figure 9. Severe edematous colitis A).... B).... C).... and D)... And put a label of A, B, C, etc. on the figures so it can be identified.
=> I leave it up to the editor to change if needed
Some of the figures had offset margins and split between pages - correct.
Line 177 nasogastric tube did not yielded any reflux..
=> changed
Line 178 .. except for one when he the horse was examined by.
=> changed
Line 185 is Table 1
=> changed
Line 221 were an extensive severe edematous colitis edema in 4 (redundant)
=> changed
Line 223 ... present in 2 horses while expensive hemorrhages appeared in 2 others. NOT expensive but extensive
=> changed
Line 230 ... present in Figures 11 et and 12.
=> changed
Line 233 in equids and describes prognostic factors to help the equine practitioner to assess disease ..
=> changed
Line 239 ... based on previous studies [1,17] Studies should be plural?
=> changed
Line 240 .. acorn hulls is in the environment...
=> changed
Line 243 .. Due to a presumptive diagnosis ...
=> changed
Line 245 .. have been performed to excluded potential bias.
=> changed
Line 255 why is there a number 2 superscript after shown... shown2 that tannins and phenolic acid ... What is that in reference to?
=> changed
Line 295 .. The main limitation ...
=> changed
Line 297 about of acorn intoxication ...
=> changed
Line 310 .. thickness of the colon wall at on ultrasonography.
=> changed
Reviewer 3 Report
Comments and Suggestions for Authors
The authors provide a retrospective examination of acorn intoxication in horses, with details about clinical signs, seasonality, data for diagnosis. They also tried to investigate which clinical signs are more related to final outcome (survival-not survival).
Despite the number of data provided, not all conclusion are supported by results. As the authors themselves state at the discussion, no research for tannin metabolites was performed (avoiding a definitive diagnosis to be obtained). Moreover no enteropathogens were looked for, so they could not be excluded as potential ethiologic factors.
Said that, the authors give a complete clinical description of the patients and focused on 4 of 6 inclusion criteria, that at my advice, give a good chance to have a likely diagnosis of acorn toxicity.
I would add some comments and suggestions:
-line 61"...pig, which seem..."
-statistical analysis: student test and chi squared test are based on comparison between two group. If I had not miss it, no division of the sample into 2 group was previously cited
-line 108: "represented"was repeated twice
-line 182: 0.05. removed comma and place dot.
-line149: do you mean that this was statistically lower?
-table 2: please add "MEAN" and specify the level of significalcy
- age plot is repeated twice
-please explain the box plot in the figure legend
- line 174: are you sure you can feel acorn hulls at rectal palpation?
or are you sure you didn't have some false negative?
- fig 8: "lived horse"? please modify
- figure legend is not displayed tin the same order as the pictures.
-line 231-245: the authors explain what are the main limitation of the study, in my opinion: the lack of a definitive diagnosis with tannin metabolites search, and the lack of the exclusion of entheropathogens ethyiology.
Author Response
-line 61"...pig, which seem..."
=> changed
-statistical analysis: student test and chi squared test are based on comparison between two group. If I had not miss it, no division of the sample into 2 group was previously cited
=> this is specified in the sentense (survivors vs non-survivors)
-line 108: "represented"was repeated twice
=> changed
-line 182: 0.05. removed comma and place dot.
=> changed
-line149: do you mean that this was statistically lower?
-table 2: please add "MEAN" and specify the level of significalcy
- age plot is repeated twice
=> changed
-please explain the box plot in the figure legend
=> changed, it was explained on the second one
- line 174: are you sure you can feel acorn hulls at rectal palpation?
or are you sure you didn't have some false negative?
yes for sure, we can find or palpate acorn hulls at rectal palpations. False negative is possible but it is difficult to state in which extend ?
- fig 8: "lived horse"? please modify
=> changed for 'still alive'
- figure legend is not displayed tin the same order as the pictures.
=> yes it is. However we add precision
-line 231-245: the authors explain what are the main limitation of the study, in my opinion: the lack of a definitive diagnosis with tannin metabolites search, and the lack of the exclusion of entheropathogens ethyiology.
=> we add this as a cause
Reviewer 4 Report
Comments and Suggestions for Authors
This article conducts a retrospective analysis of horses with acorn intoxication from 2011 to 2018. It primarily analyses the clinical symptoms, physical examination and laboratory examination results, and post-mortem findings of the cases, and correlates them with relevant factors and prognosis. The study design is straightforward and the writing is appropriate. However, there is room for improvement in the description of research methods, and the presentation of results needs further refinement. Additionally, some discussions appear somewhat forced.
Title:
1. The terms "acorn intoxication colitis" and "survival" are inconsistent with the subsequent descriptions in the text. Rephrasing and adjusting these elements for coherence is recommended.
Simple summary:
1. Only The institutional affiliation of only the first author is provided., with Please provide no indication of the affiliations of the other authors. (Line 5)
2. The objective states the intention to exclusively describe cases of acorn intoxication, while the title is "acorn intoxication colitis." It is recommended to maintain consistency between the title and the subsequent description. (Line 12-13)
3. This sentence mentioned “Results suggest that the intoxication may vary from year to year and that the number of cases seems to increase.” This trend needs to be interpreted with care, as it may be influenced by multiple factors and the sample size may be insufficient to illustrate. (Line 17)
4. The title and the previous section of the objective focus on factors related to survival, but the current mention pertains to factors associated with non-survival. To ensure consistency, it is advisable to align the content regarding survival factors throughout. (Line 21)
Abstract:
5. Similar to the issue raised in the simple summary. (Line 25-26)
6. The "P" should be in lowercase italics. (Line 31)
7. The term "clinical pathological findings" should be replaced with vocabulary related to hematological examination results, or laboratory examination results. resultssupplementary clinicopathological findings other than blood biochemical tests. (Line 34)
8. Similar to the issue raised in the simple summary. (Line 36)
Keyword:
9. It is suggested to keep the keyword specific to the article. Please revise the keywords.
Introduction:
10. The entire manuscript's citation of references is not arranged in the order of their appearance in the article. Please follow the reference format specified by the journal for revisions(Line 44)
11. The introduction contains an excessive description of the intoxication mechanism, lacking relevance to the focus of this study. The emphasis of this article is the analysis of clinical symptoms and prognostic factors in cases of acorn intoxication in equines. It is recommended to supplement additional background information specifically related to the clinical symptoms and prognosis of acorn intoxication in equine cases. (Line 50-65)
12. The term "suspected cases of acorn intoxication" should be revised to "confirmed cases of acorn intoxication" based on the context. Additionally, further clarification is needed to explicitly state the purpose of this retrospective analysis. (Line 66-69)
Materials and methods:
13. The word "post-mortem" should be written in straight. (Line 80)
14. The references for inclusion criteria of indicators should be supplemented. (Line 83-87)
15. It should be “p << 0.05”. (Line 98)
16. It is recommended to provide more information regarding the outcome such as 2-week post discharge interview or recheck examinations.
Result:
17. The total number of cases in the non-survival group, as well as the specific case numbers for the age group above 13 years and the subgroups with admission durations > 6 hours, > 12 hours, and > 24 hours, are not specified. "P" should be in lowercase italics. (Line 107-120)
18. The title and caption for Figure 1 are missing, and the size of the image should be adjusted consistently with the surrounding text. (Line 121)
19. Assessment criteria for lethargy, tachycardia, abnormal mucous membrane, etc., are lacking. "P" should be in lowercase italics. The circumstances of death are confusing.The phrase “two horses presented acute pulmonary edema and died quickly after admission despite intensive care” is irrelevant. Please specify how does pulmonary edema fit into prognostic parameterswhether the cause of death is related to the degree of poisoning. (Line 123-135.)
20. The table needs to be resized for consistency with the surrounding text, and the format of the table caption should be adjusted accordingly. (Table 1)
21. The capitalization of the title should be consistent throughout, and the table size, as well as the format of the table caption, needs to be adjusted. (Table 2)
22. The term "Figure1" has been previously mentioned in the text, please review for consistency. Ensure that the capitalization of the title is uniform throughout. (Line 163-164)
23. The capitalization of the title is consistent throughout. (Figure 2, 3, 5)
24. The capitalization of the title is consistent throughout, and the format of the results table aligns with Figure 1, 2, and 3. (Figure 4)
25. The term "Figure6" is repeated twice; please carefully review the language used. Ensure that the image size is consistent with the surrounding text, and maintain uniform capitalization in the title. (Figure 6, 7)
26. Missing units for numerical values. (Line 198)
27. Restructure the language to present a discussion of prognostic factors, creating a presentation that focuses on these elements rather than directly describing the results. (Line 213-214)
28. The word "post-mortem" should be written in straight. (Line 219)
29. The term "Figure 9" is repeated twice; please carefully review the language used. Ensure that the image size is consistent with the surrounding text, and maintain uniform capitalization in the title. (Figure 9, 10)
30. This study used t-tests and Chi-square tests to assess risk factors, but it didn't mention whether multivariate analysis was conducted to control for potential confounding factors. This might lead to the overestimation or underestimation of the importance of certain risk factors(line 123-137)
31. All the table and figure formats need to be reviewed carefully.
Discussion:
32. The word "post-mortem" should be written in straight. (Line 242-243)
33. The explanation for the correlation between intoxication cases and the annual variations in years seems somewhat arbitrarystrained and lacks persuasiveness. It is recommended to rephrase and provide references to strengthen this argumentconclusion. The discussion related to the potential implications of these observations for understanding the mechanism of intoxication and prevention strategies should be supplemented. (Line 246-249)
34. Why is the superscript number 2 added after the word "shown"? (Line 256)
35. Selecting cases only during the autumn season at the initial data screening phase and now stating that it exhibits apparent seasonality, aiding doctors in diagnosis during this period, is logically inconsistent. (Line 265-268)
36. The discussion is too limited, focusing solely on the content of this study without mentioning whether other research has also indicated that these symptoms may contribute to prognostic evaluation. (Line 284-289)
37. Desirable diagnostic criteria such as measurement of tannin metabolites have not been achieved, which should be added to the discussion of limitations. (Line295-303)
Conclusion:
1. Please streamline the language in the conclusion section, check for correct grammar usage, and ensure the relevance between the conclusion and the title, specifying whether it pertains to survival or non-survival.
Author Response
Title:
- The terms "acorn intoxication colitis" and "survival" are inconsistent with the subsequent descriptions in the text. Rephrasing and adjusting these elements for coherence is recommended.
=> we change title adding "non-survival"
Simple summary:
- Only The institutional affiliation of the first author is provided. Please provide the affiliations of the other authors. (Line 5)
=> done
- The objective states the intention to exclusively describe cases of acorn intoxication, while the title is "acorn intoxication colitis." It is recommended to maintain consistency between the title and the subsequent description. (Line 12-13)
=> we add 'colitis' (we previsously removed it to reach the word number)
- This sentence mentioned “Results suggest that the intoxication may vary from year to year and that the number of cases seems to increase.” This trend needs to be interpreted with care, as it may be influenced by multiple factors and the sample size may be insufficient to illustrate. (Line 17)
=> this is why we use the conditional "may vary"
- The title and the previous section of the objective focus on factors related to survival, but the current mention pertains to factors associated with non-survival. To ensure consistency, it is advisable to align the content regarding survival factors throughout. (Line 21)
=> done with title change
Abstract:
- Similar to the issue raised in the simple summary. (Line 25-26)
=> same answer
- The "P" should be in lowercase italics. (Line 31)
=> done
- The term "clinical pathological findings" should be replaced with vocabulary related to hematological examination results, or laboratory examination results (Line 34)
=> we respectfully disagree. The terms "clinical pathology" are worldwide widespread, with even a veterinary college of clinical pathology. Hematological is to restrictive.
- Similar to the issue raised in the simple summary. (Line 36)
=> same answer
Keyword:
- It is suggested to keep the keyword specific to the article. Please revise the keywords.
=> we do not understand this point. Keywords feel right and specific to us.
Introduction:
- The entire manuscript's citation of references is not arranged in the order of their appearance in the article. Please follow the reference format specified by the journal for revisions(Line 44)
=> references have been changed
- The introduction contains an excessive description of the intoxication mechanism, lacking relevance to the focus of this study. The emphasis of this article is the analysis of clinical symptoms and prognostic factors in cases of acorn intoxication in equines. It is recommended to supplement additional background information specifically related to the clinical symptoms and prognosis of acorn intoxication in equine cases. (Line 50-65)
=> we respectfully disagree. We do think that the mecanisms explanations are important. Moreover, we do think it is probably weird to describe clinical symptoms in introduction before naming them in the results part.
- The term "suspected cases of acorn intoxication" should be revised. Additionally, further clarification is needed to explicitly state the purpose of this retrospective analysis. (Line 66-69)
=> we do not understand this remark.
Materials and methods:
- The word "post-mortem" should be written in straight. (Line 80)
=> done
- The references for inclusion criteria of indicators should be supplemented. (Line 83-87)
=> We do not understand this remark. There are no reference to mention here. We just detail all clinical symptoms. for convenience for stasticical analysis, we decide to grade severity of mucous membrane abnormalities.
- It should be “p <05”. (Line 98)
=> done
- It is recommended to provide more information regarding the outcome such as 2-week post discharge interview or recheck examinations.
=> as a retrospective examination, we do not have sufficient information after discharger. However every horse left the hospital only when we were sure everything was normal and any horse were presented for any complications afterwards.
Result:
- The total number of cases in the non-survival group, as well as the specific case numbers for the age group above 13 years and the subgroups with admission durations > 6 hours, > 12 hours, and > 24 hours, are not specified. "P" should be in lowercase italics. (Line 107-120)
=> data have been added
- The title and caption for Figure 1 are missing, and the size of the image should be adjusted consistently with the surrounding text. (Line 121)
=> done
- "P" should be in lowercase italics. The phrase “two horses presented acute pulmonary edema and died quickly after admission despite intensive care” is irrelevant. Please specify how does pulmonary edema fit into prognostic parameters. (Line 123-135.)
=> it is state that two horses died very quickly (<3h) due to pulmonary edema. It not a questionned of pronostic parameters but just a relevant clinical information for practionner
- The table needs to be resized for consistency with the surrounding text, and the format of the table caption should be adjusted accordingly. (Table 1)
- The capitalization of the title should be consistent throughout, and the table size, as well as the format of the table caption, needs to be adjusted. (Table 2)
- The term "Figure1" has been previously mentioned in the text, please review for consistency. Ensure that the capitalization of the title is uniform throughout. (Line 163-164)
- The capitalization of the title is consistent throughout. (Figure 2, 3, 5)
- The capitalization of the title is consistent throughout, and the format of the results table aligns with Figure 1, 2, and 3. (Figure 4)
- The term "Figure6" is repeated twice; please carefully review the language used. Ensure that the image size is consistent with the surrounding text, and maintain uniform capitalization in the title. (Figure 6, 7)
=> corrected
- Missing units for numerical values. (Line 198)
=> done
- Restructure the language to present a discussion of prognostic factors, creating a presentation that focuses on these elements rather than directly describing the results. (Line 213-214)
=> part 3.5 is still a results part, not a discussion part
- The word "post-mortem" should be written in straight. (Line 219)
=> done
- The term "Figure 9" is repeated twice; please carefully review the language used. Ensure that the image size is consistent with the surrounding text, and maintain uniform capitalization in the title. (Figure 9, 10)
=> corrected
- This study used t-tests and Chi-square tests to assess risk factors, but it didn't mention whether multivariate analysis was conducted to control for potential confounding factors. This might lead to the overestimation or underestimation of the importance of certain risk factors(line 123-137)
=> in the discussion, we mentionned than we did not do multivariable analysis and that is one of our limitations
- All the table and figure formats need to be reviewed carefully.
Discussion:
- The word "post-mortem" should be written in straight. (Line 242-243)
=> done
- The explanation for the correlation between intoxication cases and the annual variations in years seems somewhat arbitrary. It is recommended to rephrase and provide references to strengthen this conclusion. The discussion related to the potential implications of these observations for understanding the mechanism of intoxication and prevention strategies should be supplemented. (Line 246-249).
=> here we just state that the annual variations is quite remarkable and propose so explanations in the discussion. But indeed, explanations for such phenomenon remain speculative with very few data/study to support one theory over another.
- Why is the superscript number 2 added after the word "shown"? (Line 256)
=> deleted
- Selecting cases only during the autumn season at the initial data screening phase and now stating that it exhibits apparent seasonality, aiding doctors in diagnosis during this period, is logically inconsistent. (Line 265-268)
=> yes we state that all cases were admit for September 19 to october 30th. The seasonality was quite impressive with most cases arriving on a couple of weeks and the rest of cases over this 40 days. Indeed inclusion criteria include fall season but fall season is much longer than 40 days!
- The discussion is too limited, focusing solely on the content of this study without mentioning whether other research has also indicated that these symptoms may contribute to prognostic evaluation. (Line 284-289)
=> as stated, there no other study on acorn intoxication in horses, neither study on pronostic factors, neither large study in others species. So it is difficult to make reference to other studies. We could have done a discussion about pronostic factors for other type of colitis or other intoxications but the subject seem us very big and general for a case series discussion. If really needed, we could make a comparison with other type of colitis but we are not sure than this article gains in quality.
- Desirable diagnostic criteria such as measurement of tannin metabolites have not been achieved, which should be added to the discussion of limitations. (Line295-303)
=> done
Conclusion:
- Please streamline the language in the conclusion section, check for correct grammar usage, and ensure the relevance between the conclusion and the title, specifying whether it pertains to survival or non-survival.
=> done
Round 2
Reviewer 4 Report
Comments and Suggestions for Authors
Simple summary and abstact:
1. The objective states the intention to exclusively describe cases of acorn intoxication, while the title is "acorn intoxication colitis." It is recommended to maintain consistency between the title and the subsequent description. (Line 12-13, Line 39)
Introduction:
4. As the title was revised to factors associated with non-survival, please adjust the “survival of 25 horses” accordingly. (Lines 67)
Materials and methods:
5. Instead of conducting definite diagnoses, this study included acorn intoxication cases based on certain criteria. It is recommended to provide specific references for this diagnosis method which is important for the credibility and traceability of the study. (lines 73-87)
Author Response
Simple summary and abstact:
- The objective states the intention to exclusively describe cases of acorn intoxication, while the title is "acorn intoxication colitis." It is recommended to maintain consistency between the title and the subsequent description. (Line 12-13, Line 39)
=> done
Introduction:
- As the title was revised to factors associated with non-survival, please adjust the “survival of 25 horses” accordingly. (Lines 67)
=> done
Materials and methods:
- Instead of conducting definite diagnoses, this study included acorn intoxication cases based on certain criteria. It is recommended to provide specific references for this diagnosis method which is important for the credibility and traceability of the study. (lines 73-87)
=> done
